# Coenzyme Q10 and Male Infertility: A Systematic Review

**DOI:** 10.3390/antiox10060874

**Published:** 2021-05-30

**Authors:** Gianmaria Salvio, Melissa Cutini, Alessandro Ciarloni, Lara Giovannini, Michele Perrone, Giancarlo Balercia

**Affiliations:** 1Division of Endocrinology and Metabolic Diseases, Umberto I Hospital, Via Conca 71, 60126 Ancona, Italy; g.salvio@pm.univpm.it (G.S.); melissa.cutini@ospedaliriuniti.marche.it (M.C.); a.ciarloni@pm.univpm.it (A.C.); lara.giovannini@ospedaliriuniti.marche.it (L.G.); m.perrone@pm.univpm.it (M.P.); 2Department of Clinical and Molecular Sciences, Polytechnic University of Marche, Via Tronto 10/A, 60126 Ancona, Italy

**Keywords:** Coenzyme Q10, antioxidants, oxidative stress, male infertility, idiopathic oligoasthenozoospermia

## Abstract

Infertility affects 15% of couples worldwide. A male factor is involved in 50% of cases. The etiology of male infertility is poorly understood, but there is evidence for a strong association between oxidative stress (OS) and poor seminal fluid quality. For this reason, therapy with antioxidants is one of the cornerstones of empirical treatment of male infertility. Coenzyme Q10 (CoQ10)—an essential cofactor for energy production with major antioxidant properties—is commonly used to support spermatogenesis in idiopathic male infertility. This systematic review aims to elucidate the usefulness of CoQ10 supplementation in the treatment of male infertility, particularly with regard to semen quality assessed by conventional and advanced methods, and pregnancy rates. All studies report a beneficial effect of CoQ10 supplementation on semen parameters, although randomized controlled trials are a minority. Moreover, the optimal dosage of CoQ10 or how it can be combined with other antioxidant molecules to maximize its effect is unknown. However, CoQ10 is still one of the most promising molecules to treat idiopathic male infertility and warrants further investigation.

## 1. Introduction

Couple infertility is defined by the WHO (World Health Organization) as the failure to initiate a pregnancy after 12 months or more of regular unprotected sexual intercourse [1]. Infertility affects about 15% of couples worldwide; in 50% a male factor, including poor sperm quality, is present. Currently, the etiology of male infertility is poorly understood. Environmental and genetic factors have been implicated, but in about one in three infertile patients a specific cause is still not identified: these patients are considered as suffering from male idiopathic infertility [2]. In recent years, a growing body of evidence has highlighted the role of oxidative stress (OS) in disorders of spermatogenesis, with 30–80% of infertile men having elevated seminal reactive oxygen species (ROS) levels [3]. ROS are oxygen—derived free radicals produced as natural byproducts of oxidative metabolic pathways and mitochondrial respiratory chain enzyme activities. In semen, ROS are mainly produced by leukocytes or abnormal or immature spermatozoa. Small quantities of ROS ensure normal cellular functions such as sperm capacitation, hyperactivation, acrosome reaction and sperm-oocyte-fusion [4]. Seminal fluid is also rich in antioxidants, which are produced in two forms: the enzymatic system (glutathione peroxidase, superoxide dismutase -SOD- and catalase -CAT) and the non-enzymatic system; the latter is composed of multiple compounds that are consumed through the diet or as supplements [5]. When ROS levels exceed the antioxidant capacity of seminal fluid, OS occurs and causes sperm DNA damage. In fact, spermatozoa are susceptible to OS because their plasma membrane contains a high percentage of polyunsaturated fatty acids that undergo lipid peroxidation by ROS. Lipid peroxidation leads to a chain reaction that generates further mutagenic and genotoxic molecules (i.e., malondialdehyde, 4-hydroxynonenal and acrolein), which in turn increase ROS production, DNA fragmentation and sperm apoptosis [4]. In sperm cells, this process is also counterbalanced by the antioxidant effect of the cytosolic selenoenzyme glutathione peroxidase 4 (Gpx4), a regulator of a specific form of non-apoptotic cell death called ferroptosis, which prevents lipid peroxidation and promotes the proper development of mature spermatozoa [5] (Figure 1). Therefore, in recent years counteracting OS through ROS reduction or antioxidant therapy has been the rationale for empiric therapy of idiopathic male infertility, although a unanimous consensus about its effectiveness is still lacking. Coenzyme Q10 (CoQ10) is an obligatory co-factor with strong antioxidant properties involved in mitochondrial energy production, which is essential in maintaining the efficient energy system of spermatozoa and protecting their membranes from lipid peroxidation. Accordingly, low CoQ10 levels have been associated with several conditions determining infertility, such as varicocele and oligozoospermia [6], and CoQ10 is one of the most widely used antioxidants in treating idiopathic male infertility [7].

The aim of the present review was to investigate the effects of CoQ10 supplementation (alone or in association with other molecules) on male fertility with emphasis on seminal parameters, oxidative stress markers, indices of sperm DNA damage and pregnancy rate.

## 2. Materials and Methods

This systematic review was performed according to the PRISMA statement [8].

### 2.1. Literature Search

All studies were identified in the PubMed and SCOPUS databases between February and April 2021. The following search terms were used: “Q10”, “Q(10)”, “Coenzyme Q”, “CoQ”, “CoQ10”, “Ubiquinone”, “Ubiquinol”, “Infertility”, “Male infertility”, “Male fertility”, “Testosterone”, “Seminal” and “Sperm parameters” in various combinations. Moreover, bibliographic references of articles included in the first screening were reviewed for any additional relevant articles.

### 2.2. Inclusion Criteria

Papers were considered eligible if they met the following criteria:Studies evaluating the effect of CoQ10 or ubiquinol (a reduced form of CoQ10) supplementation (alone or with other antioxidant molecules) on male fertility

### 2.3. Exclusion Criteria

Animal studiesScientific articles that were not published in EnglishReview or conference abstracts or letters to the editorStudies published before 2000

For duplicate studies, we included only the article with more detailed data.

### 2.4. Selection of Studies

Three reviewers (A.C., M.P. and G.S.) independently evaluated the title, abstract and full text of each potentially relevant manuscript for eligibility. Any discrepancies were resolved by discussion. The following data were extracted from all qualified studies: authors, year of publication, study design, sample size, age of patients, target population, type and dosage of antioxidants, length of therapy and results.

## 3. Results

The search strategy matched 486 articles; 449 were excluded on the basis of the title and/or abstract. Of the remaining 37, 24 matched the inclusion and exclusion criteria (Figure 2). Twelve studies used CoQ10 (or ubiquinol) alone (Table 1), whereas the other 12 studies evaluated the effect of compound combinations (Table 2). All the studies were prospective except one [9]; 7 studies had a randomized, placebo-controlled and double-blind design. Daily CoQ10 dosage ranged from 20 mg [10,11,12] to 400 mg [13].

### 3.1. CoQ10 and Male Fertility: Monotherapy

The earliest study evaluating the effect of CoQ10 supplementation on seminal parameters was the pilot study by Balercia and co-workers (2004), which included 22 subjects with idiopathic asthenozoospermia who underwent supplementation with CoQ10 200 mg/day twice a day. After 6 months, a significant difference was found in the forward (class a + b) motility of sperm cells (from 9.13 ± 2.50 to 16.34 ± 3.43%, *p* < 0.05), confirmed by computer-assisted sperm analysis (CASA) determination of curvilinear velocity (from 26.31 ± 1.50 to 46.43 ± 2.28 µm/sec, *p* < 0.05) and straight progressive velocity (from 15.20 ± 1.30 to 20.40 ± 2.17 µm/s, *p* < 0.05). At the same time, CoQ10 levels increased about 3-fold in seminal plasma (from 42.0 ± 5.1 to 127.1 ± 1.9 ng/mL, *p* < 0.05) and 2-fold in sperm cells (from 3.1 ± 0.4 to 6.5 ± 0.3 ng/10^6^ cells, *p* < 0.005), supporting a cause-effect relationship between sperm motility and seminal CoQ10 concentrations. After a 6-months period of washout, seminal parameters reverted to levels comparable to those measured at baseline [14]. In a subsequent randomized, double-blind, placebo-controlled study, the same group studied the effect of daily administration of 200 mg of CoQ10 for 6 months in 28 subjects and 27 control subjects. Again, coQ10 levels in seminal fluid significantly increased after treatment in both plasma (from 61.29 ± 20.24 to 99.39 ± 31.51 ng/mL, *p* < 0.0001) and sperm cells (from 2.44 ± 0.97 to 4.57 ± 2.46 ng/10^6^ cells, *p* < 0.0001). Likewise, the authors observed an increase in sperm cells total and forward motility (from 33.14 ± 7.12 to 39.41 ± 6.80, *p* < 0.0001), and (from 10.43 ± 3.52 to 15.11 ± 7.34%, *p* < 0.0003, respectively), again confirmed by CASA. Moreover, ubiquinol levels in seminal plasma and sperm cells also increased after treatment [15]. 

In 2009, Safarinejad evaluated the effect of CoQ10 on semen parameters in 212 infertile men with idiopathic oligoasthenoteratospermia (iOAT). After randomization, 106 subjects underwent a 26-week treatment with 300 mg CoQ10 once daily and 106 subjects received matching placebo. After treatment, statistically significant differences were found between the groups, with higher total sperm counts (47.8 ± 11.2 × 10^6^ vs. 57.6 ± 14.4 × 10^6^, *p* = 0.01) and motility values (23.1 ± 2.1% vs. 27.6 ± 2.2%, *p* = 0.01) in the coQ10 group. Interestingly, the coQ10 group showed a significant decrease in FSH and LH levels and a significant increase in inhibin B levels. In particular, serum FSH decreased by 35.9% (*p* = 0.03) and serum inhibin B increased by 22.1% (*p* = 0.03), suggesting a direct effect on the seminiferous tubules. A significant increase in the percentage of acrosome reaction was also observed in the CoQ10 group but not in the placebo group. Moreover, a significant increase was observed in blood and seminal plasma CoQ10, with strong and significant correlations between seminal plasma CoQ10 and sperm count (r = 0.77, *p* = 0.01), sperm motility (r = 0.76, *p* = 0.01) and sperm morphology (r = 0.54, *p* = 0.02). After a 56-week free-treatment phase, semen parameters were still different between the groups, albeit the difference was not statistically significant [16]. 

A randomized, double-blind, placebo-controlled trial found no significant differences in sperm parameters after 3 months of 200 mg/day CoQ10 administration, but it described a significant increase in total antioxidant capacity of seminal plasma (4.11 ± 0.63 μmol/L and 4.45 ± 0.36 μmol/L, *p* = 0.017 after treatment, respectively) and a significant reduction in serum malondialdeyde (3.59 ± 0.4 μmol/L and 3.42 ± 0.34 μmol/L, *p* = 0.013, before and after treatment, respectively), both indicative of a positive effect on oxidative stress markers [17].

In 2012, Safarinejad et al. enrolled 228 patients with iOAT in a randomized, double-blind, placebo-controlled study. Patients were randomly assigned 1:1 to 2 groups to receive 200 mg ubiquinol daily or placebo for 26 weeks. At the end of the treatment phase, significant differences in sperm density (16.8 ± 4.4 × 10^6^/mL vs. 28.7 ± 4.6 × 10^6^/mL, *p* = 0.005), sperm motility (25.4 ± 2.1% vs. 35.8 ± 2.7%, *p* = 0.008) and sperm morphology (14.8 ± 4.1% vs. 17.6 ± 4.4%, *p* = 0.01) were observed in the ubiquinol group compared to the placebo group. Moreover, ubiquinol treatment induced a significant reduction from baseline in serum LH (33.9%, *p* = 0.03) and FSH (37.7%, *p* = 0.02) and a significant increase in serum inhibin B levels (25.2%, *p* = 0.01). A slight and statistically nonsignificant increase in serum testosterone was also observed (19.5%, *p* = 0.08). No significant changes in hormonal levels were observed in the placebo group. Positive correlations were observed between semen parameters and duration of the treatment with ubiquinol and between seminal plasma antioxidant capacity (assessed by measuring catalase-like and superoxide dismutase-like activity) and semen parameters. After a 12-week treatment-free period, the improvement in semen parameters gradually reversed, but the difference between the groups was still significant for sperm density and sperm motility. Serum FSH concentrations remained significantly lower [18]. 

In the same year, Safarinejad et al. conducted an open-label prospective study on 287 iOAT patients who received CoQ10 300 mg daily for 12 months and were followed up for another 12 months after CoQ10 discontinuation to assess the effect of CoQ10 on pregnancy rates. Semen parameters significantly improved after 3 months of treatment and reached a peak at 12 months. After CoQ10 discontinuation, they gradually decreased, but were still significantly higher than baseline at 24-month follow-up. Similarly, serum FSH and LH decreased during treatment and gradually increased after CoQ10 discontinuation, but FSH remained 38% lower (*p* = 0.01) and LH remained 25% lower (*p* = 0.02) than baseline. In contrast, inhibin B increased during treatment and declined after discontinuation, remaining significantly higher than baseline at the end of the 24-month study period (*p* = 0.01). The total pregnancy rate at a mean time of 8.4 ± 4.7 months was 34.1%. All three semen parameters showed a linear association with the pregnancy rate [19].

Festa et al. investigated the effect of 100 mg CoQ10 daily supplementation on semen parameters of 38 patients with varicocele-related male infertility. After a 12-week period of treatment, they measured a significant increase in sperm density (35.5 ± 3.4 × 10^6^/mL vs. 42.6 ± 4.5 × 10^6^/mL, *p* = 0.03) and forward motility (20.1 ± 4.5% vs. 28.4 ± 4.9, *p* = 0.03). Seminal plasma total antioxidant capacity (TAC) also increased significantly after treatment (106.6 ± 8.7 s vs. 148.4 ± 12.6 s, *p* < 0.01) [20].

In a randomized, double-blind, placebo-controlled trial, Nadjarzadeh et al. explored the effects of 3-month CoQ10 supplementation on antioxidant enzymes and isoprostane concentration in seminal plasma and their relationship with seminal CoQ10 concentration in a group of men with iOAT. After CoQ10 administration, forward and total motility, but not the other parameters, showed an increase. CoQ10 significantly increased after supplementation (44.74 ± 36.47 ng/mL vs. 68.17 ± 42.41 ng/mL, *p* = 0.0001), showing a positive correlation with normal sperm morphology (r = 0.44, *p* = 0.037). Men in the CoQ10 group had higher catalase and SOD activity and lower seminal plasma 8-isoprostane concentration than those in the placebo group after the treatment period. Moreover, seminal CoQ10 showed a significant correlation with SOD (r = 0.6, *p* < 0.005) and catalase (r = 0.3, *p* < 0.05) after treatment [21].

The effect of 200 mg ubiquinol supplementation was evaluated in a retrospective study of 62 infertile normo- or mild oligospermic patients with asthenoteratozoospermia. After 6 months of treatment, normal morphology and sperm motility (a + b) showed a significant increase (*p* < 0.001), whereas sperm concentration did not [9].

In a prospective uncontrolled study published in 2015, 60 oligospermic patients received 150 mg ubiquinol daily supplementation for 6 months. After the treatment, the total sperm count increased by 53% (*p* < 0.05) and total sperm motility increased by 26% (*p* < 0.05), whereas the number of immobile or sluggish motile sperm fell to 55% (*p* < 0.05) and 29% (*p* < 0.05), respectively [22].

Sixty-five patients with iAOT have recently been recruited in a prospective randomized clinical trial by Alahmar, who compared the effect of 2 different dosages of CoQ10 (200 and 400 mg) on semen parameters and oxidative stress markers. Treatment with CoQ10 for 3 months resulted in a significant increase in sperm concentration, progressive motility and total motility in both groups, but changes in the kinetic parameters were greater in the 400 mg CoQ10 group. Similarly, SOD and CAT activity increased after treatment, but changes were still greater in subjects treated with 400 mg CoQ10. TAC, CAT and SOD activity increased in both groups (although the TAC increase was not significant in subjects treated with 200 mg CoQ10) and showed a significant positive correlation with semen parameters in both groups [13].

Seventy infertile men with iOAT were randomized by Alahmar et al. to receive CoQ10 (200 mg/day) or selenium (200 mcg/day) for 3 months. The CoQ10 group exhibited a significant increase in sperm concentration (8.22 ± 6.88 × 10^6^/mL vs. 12.53 ± 8.11 × 10^6^/mL, *p* < 0.01), progressive sperm motility (16.54 ± 9.26% vs. 22.58 ± 10.15%, *p* < 0.01) and total sperm motility (25.68 ± 6.41% vs. 29.96 ± 8.09%, *p* < 0.01). There was also a significant improvement in TAC (1.1 ± 0.30 mmol/L vs. 1.28 ± 0.26 mmol/L, *p* < 0.01), SOD (12.6 ± 3.71 U/mL vs. 15.4 ± 4.31 U/mL, *p* < 0.01) and CAT (11.3 ± 2.53 U/mL vs. 12.5 ± 2.24 U/mL, *p* < 0.05). Similar results, except for sperm concentration and CAT, were observed in the selenium group, although changes were lower [23].

The above data are summarized in Table 1.

### 3.2. Compound Combinations and Male Infertility

In the prospective open-label study of Busetto et al., 114 patients with iOAT received a formulation containing multiple antioxidant molecules (L-carnitine, acetyl-L-carnitine, fructose, citric acid, selenium, zinc, ascorbic acid, cyano-cobalamin, folic acid) and 20 mg CoQ10 once daily for 4 months. After treatment, sperm motility increased significantly from 18.3 ± 3.8 to 42.1 ± 5.5 (*p* < 0.05), whereas no significant changes were observed in sperm density and sperm morphology [10].

Abad et al. in a prospective study examined 20 infertile men with asthenoteratozoospermia who received antioxidant treatment with commercial multi-vitamins containing 20 mg coQ10, 1500 mg L-carnitine, 60 mg vitamin C, 10 mg vitamin E, 200 mcg vitamin B9, 1 mcg vitamin B12, 10 mg zinc and 50 mcg selenium for 3 months. A slight increase was observed after treatment in sperm density (69.75 ± 44.08 × 10^6^/mL vs. 69.85 ± 50.55 × 10^6^/mL, *p* = 0.042) and normal morphology (4.10 ± 3.21% vs. 5.57 ± 5.64%, *p* = 0.04), whereas type A motility, type A + B motility and vitality exhibited a marked and significant improvement. No changes in total sperm and type B motility were observed. The progression of sperm DNA fragmentation over increasing incubation time (0 h, 2 h, 6 h, 8 h and 24 h) showed a significant reduction at all time points and percentage of DNA degraded sperm was significantly reduced (7.32 ± 4.12% vs. 5.66 ± 3.21%, *p* = 0.04) after antioxidant treatment. The ongoing pregnancy rate, calculated 3 months after treatment by telephone interview, was 15% [11].

Kobori et al. investigated the effects of antioxidant therapy with 120 mg CoQ10, 80 mg vitamin C and 40 mg vitamin E daily on the semen parameters of 169 men with iOAT. Semen was collected before and 3 and 6 months into treatment. Pregnancy outcome was also evaluated at the same time points. A significant increase was observed in sperm concentration at 3 (*p* = 0.03) and 6 months (*p* < 0.001). Similarly, sperm motility showed a significant increase at 3 (*p* < 0.001) and 6 months (*p* < 0.001). A total of 48 (28.4%) pregnancies, including 16 spontaneous pregnancies, had been achieved at last follow-up [24]. 

The antioxidant capacity of a combination of CoQ10 (200 mg) and aspartic acid (2660 mg) was investigated in the observational study by Tirabassi et al., who enrolled 20 patients with idiopathic asthenozoospermia. After a 3-month treatment period, there was a significant improvement in sperm kinetics, but not in sperm count or in the number of atypical sperm cells. Levels of nitric oxide and peroxynitrite in seminal plasma decreased, whereas SOD activity increased. Moreover, the percentage of damaged DNA (assessed by comet assay) decreased significantly. A negative correlation was calculated between the increase of CoQ10 and the decrease of nitric oxide (r = −0.755, *p* < 0.001) and between the increase of CoQ10 and the decrease of DNA damage (r = −0.496, *p* = 0.026), whereas a positive correlation was described between the increase of CoQ10 and the increase of SOD activity (r = 0.679, *p* < 0.001). Changes in SOD activity and in nitric oxide levels correlated with the decrease of DNA damage index (r = −0.456, *p* = 0.044 and r = 0.458, *p* = 0.042, respectively) [25].

Gvozdjáková et al. investigated the effect of antioxidant treatment with a multi-vitamin complex (ubiquinol, carnitine, vitamin E and vitamin C) in a sample of 40 infertile men with oligoasthenoteratozoospermia aged 28–40 years. Data were collected at baseline and at 3 and 6 months. Sperm density significantly increased after 3 (by 39.8%, *p* < 0.001) and 6 months (by 78.0%, *p* < 0.001) of treatment. CoQ10 and α-tocopherol concentrations increased significantly whereas the levels of thiobarbituric acid reactive substances (an oxidative stress parameter) decreased significantly [26].

In a prospective, open-label, non-randomized study, Lipovac et al. compared the effect of a combination of micronutrients (including 15 mg CoQ10 and carnitine) vs. carnitine alone in infertile patients with at least one abnormal semen analysis. After 3 months, all studied sperm parameters significantly improved, but the relative change of sperm density and progressive motility was higher in the combined micronutrient treatment group [27].

In a recent double-blind, placebo-controlled trial, 77 infertile men with a high DNA fragmentation index (≥25%) were randomized to receive a commercial fertility supplement containing vitamins and antioxidants (including 10 mg CoQ10) or placebo twice a day for 6 months. The antioxidant group had higher sperm density after 3 months (median: 24.4 × 10^6^/mL vs. 27.2 × 10^6^/mL, *p* = 0.028) compared to pre-treatment values. The DNA fragmentation index did not change during the 6 months of antioxidant therapy. No significant difference between the antioxidant and the placebo group was seen for any of the semen parameters or for sperm DNA fragmentation index (DFI) after treatment [12]. 

Recently, 31 patients with idiopathic male infertility were randomized to receive an antioxidant supplement containing CoQ10 (90.26 mg), L-carnitine, zinc, astaxanthin, vitamin C, vitamin B12 and vitamin E or hochu-ekki-to (a Chinese herbal medicine). Sperm analysis and LH, FSH and testosterone serum concentrations were performed before and after 3 months of treatment. Neither the endocrinological findings nor the semen parameters significantly increased after treatment in the hochu-ekki-to group. Total motile count was the only semen parameter to show a significant increase after treatment in the supplement group (10.3 ± 8.5 × 10^6^ vs. 24.1 ± 21.9 × 10^6^, *p* = 0.04), whereas no significant changes in hormonal parameters were observed [28].

Arafa et al. have recently evaluated the effect of antioxidant supplementation on conventional semen parameters and advanced sperm function tests in a population of infertile men. On the basis of semen analysis, subjects were categorized into idiopathic (at least one abnormal sperm parameter) and unexplained (normal semen analysis) male infertility. A total of 148 subjects (119 in the idiopathic male infertility and 29 in the unexplained infertility group) received a 3-month treatment with 3 capsules twice a day of an antioxidant formula, which provided, among others, 200 mg CoQ10 per day. Sperm DNA fragmentation and oxidative stress assessment (the latter calculated through seminal oxidation-reduction potential, ORP) allowed a further classification into high or low SDF (>30% or ≤30%, respectively) and high or low ORP (>1.34 mV/10^6^ sperm/mL or ≤1.34 mV/10^6^ sperm/mL, respectively). In the idiopathic infertility group, supplementation showed a significant decrease in seminal ORP and SDF levels. Sperm analysis revealed an improvement in all parameters investigated, except for semen volume and sperm viability. Changes were also more evident in the high ORP and high SDF sub-categories. Moreover, ORP levels decreased significantly in all the subjects with idiopathic infertility and in the high ORP and high SDF sub-categories. In the unexplained male infertility group, only progressive motility improved significantly. In the idiopathic infertility group, ORP levels decreased after treatment [29].

In a recent single-blinded trial, Sadaghiani et al. enrolled 50 oligospermic and asthenospermic subjects who were also active smokers to investigate the effect of antioxidant supplementation (30 mg CoQ10, 8 mg zinc, 100 mg vitamin C, 12 mg vitamin E, 400 μg folic acid once a day and 200 mg selenium every other day) on semen parameters. After 3 months, the mean volume of sperm rose from 3.48 ± 1.44 to 3.71 ± 1.42 mL (*p* = 0.032), the total sperm count rose from 21.76 ± 23.02 to 23.22 ± 23.28 × 10^6^ (*p* = 0.001), total sperm motility changed from 27.22 ± 13.69 to 31.85 ± 5.82% (*p* = 0.001); progressive motility increased from 9.82 ± 9.10 to 11.57 ± 10.18% (*p* = 0.001), and normal sperm morphology changed from 23.22 ± 23.28 to 33.60 ± 20.01% (*p* = 0.003) [30]. 

In the randomized, double-blind, placebo-controlled study by Kopets et al., 83 patients with idiopathic male infertility were enrolled to receive a dietary supplement containing a mix of antioxidants and vitamins (including 40 mg CoQ10) or placebo (1:1) for 6 months. The primary outcome was normalization of semen analysis at 0, 2 and 4 months, whereas the secondary outcome was the pregnancy rate. At 4 months, 69.0% of the patients in the treatment group and 22.0% in the placebo group had normal semen analysis (*p* < 0.001), whereas the pregnancy rate was significantly higher in the treatment than the placebo group (23.8% and 4.9%, *p* = 0.017) [31]. 

In a recent open study by Nazari et al., 180 male patients with iOAT received an antioxidant mix including 40 mg CoQ10 daily for 3 months. A significant increase in sperm density (25 vs. 36 × 10^6^/mL, *p* = 0.004) and morphology (*p* = 0.01), but not sperm motility (*p* = 0.2) was observed after treatment [32].

The above data are summarized in Table 2.

## 4. Discussion

There is mounting experimental evidence for a role of OS in idiopathic male infertility [6]. The term Male Oxidative Stress Infertility (MOSI) has recently been proposed as a novel descriptor for infertility with abnormal semen characteristics and OS, and it may be present in about 56 million men worldwide [3]. In a recent systematic review, Majzoub et al. reported a beneficial effect of antioxidant therapy in reversing OS-induced sperm dysfunction, but the heterogeneous nature of the study designs prevented the recognition of an optimal treatment regimen [5]. CoQ10 supplementation has demonstrated a beneficial effect on inflammatory and OS-related conditions, such as cardiac failure [33], coronary artery disease [34] and type 2 diabetes mellitus [35]. Moreover, an improvement in the serum levels of inflammatory markers has been reported in a recent meta-analysis by Zhang et al. [36]. 

Lafuente et al. in 2013 conducted a meta-analysis to evaluate the effect of CoQ10 treatment on live birth, pregnancy rate, level of Q10 in seminal plasma, sperm concentration and total sperm motility. Three randomized, double-blind, placebo-controlled trials [15,16,17] were included in the analysis, with 149 men in the CoQ10 group and 147 men in the placebo group. The result of the meta-analysis showed a significant increase in sperm concentration, sperm motility and seminal CoQ10 concentration, whereas no significant effects were observed on pregnancy rate. Moreover, none of the studies provided any data regarding live birth [37].

The present review demonstrated a general positive effect of CoQ10 supplementation on seminal parameters. As regards CoQ10 monotherapy, sperm motility significantly increased in all the studies evaluated [9,13,14,15,16,18,19,20,21,22,23] except one [17]. A significant increase in sperm concentration was also reported by some authors [13,16,18,19,20,23], whereas the effects on normal morphology were lower [9,13,18,19,23]. On the other hand, supplementation with compound mixes (including 20–200 mg CoQ10) showed similar effects on sperm density [11,12,24,26,27,29,31,32], motility [10,11,24,25,27,29,30,31] and morphology [11,29,30,31,32]. However, the extreme variability of the CoQ10 dosage used in mixed compounds (20–200 mg per day) and the lack of a direct comparison between different nutraceutical preparations makes it difficult to understand how much CoQ10 affects seminal parameters compared to other molecules in mixed compounds. There are no studies comparing CoQ10 supplementation versus combinations of antioxidants. However, compared to selenium supplementation, CoQ10 supplementation seems to provide better results both in terms of seminal parameters and of antioxidant capacity of seminal fluid [23]. 

As noted above, spermatozoa are susceptible to OS that is generated when seminal fluid scavenging mechanisms are overwhelmed by ROS [4]. CoQ10 supplementation significantly increases seminal coQ10 levels [14,15] and improves the antioxidant capacity of seminal fluid [13,17,18,20,21,23], improving both enzymatic and non-enzymatic germ cell protection systems. This appears critical in protecting sperm DNA from ROS damage, as demonstrated by some authors [11,25,28,29], who reported a significant reduction in the DFI after antioxidant treatment. DFI is commonly used to evaluate sperm chromatin integrity and is increasingly being used for its ability to diagnose male fertility potential in couples seeking medical assistance with assisted reproductive technology [38]. DFI improvement provides further evidence of the usefulness of antioxidant therapy in male infertility, but further work is required.

The positive effect of CoQ10 supplementation on spermatogenesis has also been confirmed by the reduction of FSH levels and the increase of inhibin B levels [16,18,19]. Inhibin B is produced by Sertoli cells and its serum levels strongly correlate with testicular volume and sperm counts. It controls FSH secretion via a negative feedback, so the levels of these hormones are inversely proportional [39]. Thus, an increase in inhibin B associated with a reduction in FSH appears strongly indicative of an improvement in testicular function. In any case, the beneficial effects on semen parameters seem to decrease after treatment discontinuation [14,16,18,19]

In most studies, the optimal dosage of coenzyme CoQ10 was considered 200 mg/day, based on the first study by Balercia et al. [15], but favorable results have also been obtained with lower dosages [20,22]. On the other hand, the only study evaluating two different treatment regimens showed a greater improvement in seminal parameters in the group treated with 400 mg CoQ10 than in the group treated with 200 mg CoQ10 [13]. Therefore, more data are needed to determine the best dosage of CoQ10.

Finally, it should be noted that only 5 of 12 studies (42%) with CoQ10 and 2 of 12 studies (12%) with combination treatment had a randomized controlled design, while data on pregnancy rates remain sparse or anecdotal [19,31]. These are important limitations. First, studies with uncontrolled design do not allow excluding the influence of lifestyle modifications on semen quality. Indeed, nutritional factors such as a high-fat and/or a high-energy diet have a negative effect on seminal parameters and seminal OS indices, whereas high fruit, vegetable, fish and legume intakes are strongly associated with better seminal parameters [40]. Consequently, the lack of a control group does not allow adequate differentiation between the effect of CoQ10 supplementation and that of a healthier lifestyle. Moreover, antioxidant supplementation should not be considered as an alternative, but rather as a complement to a path of lifestyle improvement that has overall human health as its ultimate goal. On the other hand, the aim of infertility treatment is to achieve pregnancy. In the WHO 2010 manual, normal seminal parameters are established with reference to a fertile population with a time-to-pregnancy of less than 12 months; they should be used as a guide, although a direct correlation between seminal parameters and the probability of pregnancy has not been demonstrated. Moreover, since the probability of achieving a pregnancy depends on numerous factors besides semen quality, all infertility risk factors in the couple should be taken into account before recommending antioxidant supplementation. Further studies are clearly needed to determine whether CoQ10 has a merely “cosmetic” effect or whether it can have a real impact on the treatment of male infertility.

## 5. Conclusions

The qualitative analysis of available studies has shown that supplementation with CoQ10, alone or in combination with other antioxidant molecules, has a beneficial effect on seminal quality, especially regarding sperm motility. Indirect indications derive from an improvement in the antioxidant capacity of the seminal fluid and the chromatin integrity of spermatozoa. Improvements in semen parameters begin after 3–6 months of treatment but disappear when supplementation is discontinued. Further studies are needed to establish the optimal CoQ10 dosage and the possible superiority of co-administration with other molecules compared to monotherapy. Most importantly, well-structured studies are required to determine the impact of CoQ10 supplementation on pregnancy rate and live birth rate.

## Figures and Tables

**Figure 1 antioxidants-10-00874-f001:**
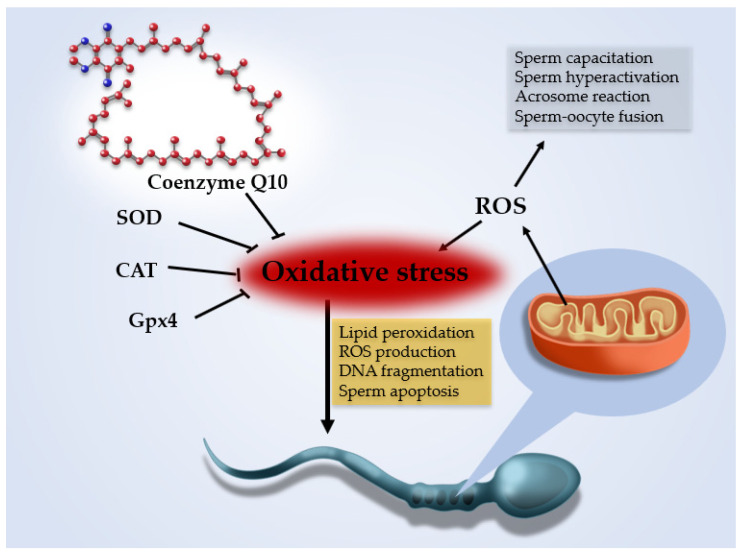
Reactive oxygen species (ROS) are oxygen-derived free radicals produced by oxidative metabolic pathways (chiefly the mitochondrial respiratory chain). Small amounts of ROS are needed to ensure normal sperm cell functions such as capacitation, hyperactivation, acrosome reaction and sperm-oocyte fusion, but excess ROS production induces oxidative stress (OS). Seminal fluid and sperm cells are rich in antioxidant molecules (superoxide dismutase -SOD-, catalase -CAT- and glutathione peroxidase 4—Gpx4), which counterbalance the effect of ROS. Spermatozoa are susceptible to OS because their plasma membrane is rich in polyunsaturated fatty acids that undergo lipid peroxidation, which in turn increases ROS production and induces DNA fragmentation and sperm apoptosis. Coenzyme Q10 is an obligatory co-factor with strong antioxidant properties that counteract OS by reducing ROS production in mitochondria and protect spermatozoa membranes from lipid peroxidation.

**Figure 2 antioxidants-10-00874-f002:**
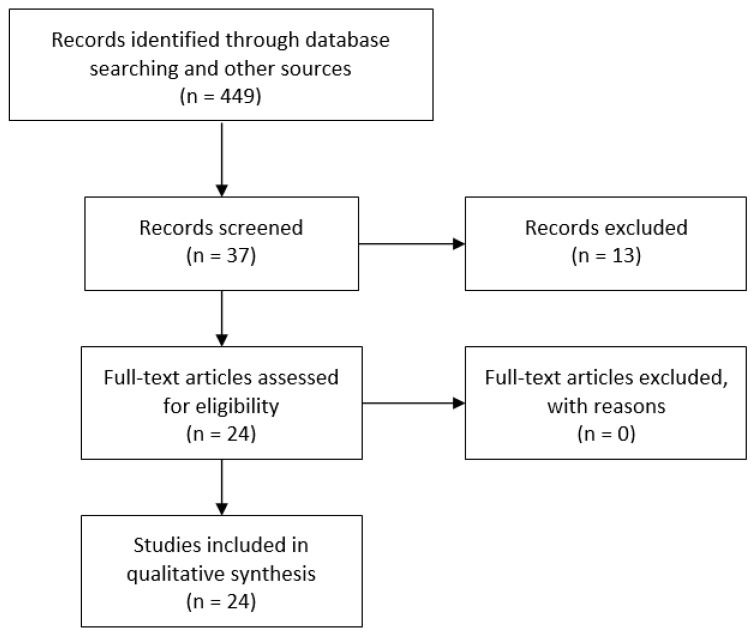
PRIMA flow diagram of literature search.

**Table 1 antioxidants-10-00874-t001:** Coenzyme Q10 and male fertility (monotherapy), characteristics of suitable studies.

Author(s)	Design	Sample Size	Age	Target Population	Daily Dosage	Consumption	Length of Follow-up	Results
Balercia et al., 2004 [14]	Open, uncontrolled	22	31 (25–39)	Idiopathic asthenozoospermia	CoQ10 200 mg/day	Twice daily orally	6 months	Imrprovement in the forward (a + b) motility of sperm cells (*p* < 0.05), improvement in computer-assisted determination of kinetic parameters (*p* < 0.05) and increase in seminal coQ10 levels (*p* < 0.05) after treatment.
Balercia et al., 2009 [15]	Randomized, double-blind, placebo-controlled	60	32 (27–39)	Idiopathic asthenozoospermia	CoQ10 200 mg/day	Twice daily orally	6 months	Improvement in the forward (*p* < 0.0001) and total (*p* < 0.0003) motility of sperm cells, improvement in computer-assisted determination of kinetic parameters and increase in seminal coQ10 and ubiquinol levels (*p* < 0.0001) after treatment. Lower baseline value of motility and levels of coQ10 had higher probability to be responders to the treatment.
Safarinejad, 2009 [16]	Randomized, double-blind, placebo-controlled	212	28 (21–42)	iOAT	CoQ10 300 mg/day	Once daily orally	26 weeks	Improvement in sperm density and motility (*p* = 0.01) after coQ10 treatment. Positive correlation between treatment duration and sperm count (*p* = 0.01), sperm motility (*p* = 0.01) and sperm morphology (*p* = 0.02). Decrease in FSH and LH levels (*p* = 0.03) and increase in inhibin levels and acrosome reaction (*p* = 0.03) after coQ10 treatment.
Nadjarzadeh et al., 2011 [17]	Randomized, double-blind, placebo-controlled	60	34 (25–46)	iOAT	CoQ10 200 mg/day	-	3 months	Non-significant changes in semen parameters of CoQ10 group. Increase in total antioxidant capacity of seminal plasma (*p* = 0.017) and significant reduction in concentration of thiobarbituric acid-reactive substances (*p* = 0.013) in the CoQ10 group.
Safarinejad et al., 2012 [18]	Randomized, double-blind, placebo-controlled	228	25–44	iOAT	Ubiquinol 200 mg/day	Once daily orally	26 weeks	Improvement in sperm density (*p* = 0.005), motility (*p* = 0.008) and morphology (*p* = 0.01) after ubiquinol treatment. Positive correlation between treatment duration, seminal plasma antioxidant capacity and semen parameters. Decrease in FSH (*p* = 0.02) and LH levels (*p* = 0.03) and increase in inhibin levels (*p* = 0.01) after ubiquinol treatment.
Safarinejad et al., 2012 [19]	Open, uncontrolled	287	32 (26–43)	iOAT	CoQ10 300 mg/day	Twice daily orally	12 months	Improvement in sperm density, motility and morphology after CoQ10 treatment (*p* < 0.05). Decrease in FSH and LH levels and increase in inhibin levels after CoQ10 treatment (*p* < 0.05). Improvements remained significant 12 months after CoQ10 discontinuation. Beneficial effect on pregnancy rate.
Festa et al., 2014 [20]	Open, uncontrolled	38	19–40	Varicocele-related infertility	CoQ10 100 mg/day	Twice daily orally	3 months	Increase in sperm density (*p* = 0.03), forward motility (*p* = 0.03) and seminal plasma total antioxidant capacity (*p* < 0.01).
Nadjarzadeh et al., 2014 [21]	Randomized, double-blind, placebo-controlled	60	34 (25–40)	iOAT	CoQ10 200 mg/day	Twice daily orally	3 months	Increase in forward and total motility (*p* < 0.05) in the treatment group. Increase in seminal CoQ10 (*p* = 0.0001) and positive correlation with normal sperm morphology (*p* = 0.037) and CAT (*p* < 0.05) and SOD activity (*p* < 0.05) after treatment. Higher catalase and SOD activity and lower seminal 8-isoprostane concentration (*p* < 0.05) in the CoQ10 group after treatment.
Cakiroglu et al., 2014 [9]	Retrospective	62	32 (23–50)	Normo- or mild oligospermic patients with asthenoteratozoospermia	Ubiquinol 200 mg/day	Twice daily orally	6 months	Increase in normal morphology and sperm motility (a + b) (*p* < 0.001) after ubiquinol supplementation.
Thakur et al., 2015 [22]	Open, uncontrolled	60	20-40	Oligospermia	Ubiquinol 150 mg/day	Once daily orally	6 months	Increase in total sperm count and total sperm motility (*p* < 0.05) and reduction of number of immobile and sluggish motile sperm (*p* < 0.05).
Alahmar, 2019 [13]	Prospective, randomized	65	27	iOAT	CoQ10 200 mg/day versus CoQ10 400 mg/day	Once daily orally	3 months	Increase in seminal parameters (sperm concentration, sperm motility and sperm morphology) and seminal antioxidant activity in both groups after treatment (*p* < 0.05). Higher increase in kinetic parameters in subjects treated with 400 mg CoQ10. Correlation between sperm parameters and seminal antioxidant activity after treatment (*p* < 0.05).
Alahmar et al., 2021 [23]	Prospective, randomized	70	25	iOAT	CoQ10 200 mg/day Versus Selenium 200 mcg/day	Once daily orally	3 months	Increase in sperm density, total sperm motility and progressive sperm motility (*p* < 0.01) and improvement of antioxidant capacity of seminal fluid (*p* < 0.05) after CoQ10 administration.

CoQ10 = coenzyme Q10; iOAT = idiopathic oligoasthenozoospermia; SOD = superoxide dismutase.

**Table 2 antioxidants-10-00874-t002:** Coenzyme Q10 and male fertility (mixed compounds), characteristics of suitable studies.

Author(s)	Design	Sample Size	Age	Target Population	Antioxidant	Consumption	Length of Follow-up	Results
Busetto et al., 2012 [10]	Open, uncontrolled	114	31 (21–46)	iOAT	CoQ10 20 mg L-carnitine 145 mg Acetyl-L-carnitine 64 mg Fructose 250 mg Citric acid 50 mg Selenium 50 mcg Zinc 10 mg Ascorbic acid 90 mg Cyanocobalamin 1.5 mcg Folic acid 200 mcg	Once daily orally	4 months	Increase in progressive sperm motility (*p* < 0.05).
Abad et al., 2013 [11]	Open, uncontrolled	20	-	Asthenoteratozoospermia	CoQ10 20 mg L-carinitine 1500 mg vitamin C 60 mg vitamin E 10 mg vitamin B9 200 mcg vitamin B12 1 mcg Zinc 10 mg selenium 50 mcg	-	3 months	Slight increase in sperm density (*p* = 0.042) and normal morphology (*p* = 0.04), pronounced increase in A motility, A + B motility and vitality (*p* < 0.05) after antioxidant treatment. Improvement in DNA integrity and reduction in proportion of highly DNA degraded sperm (*p* = 0.04).
Kobori et al., 2014 [24]	Open, uncontrolled	169	36 (25–58)	iOAT	CoQ10 120 mg Vitamin C 80 mg Vitamin E 40 mg	Twice daily orally	6 months	Increase in sperm concentration and sperm motility at 3 and 6 months of treatment (*p* < 0.05).
Tirabassi et al., 2015 [25]	Open, uncontrolled	20	32	Idiopathic asthenozoospermia	CoQ10 200 mg Aspartic acid 2660 mg	Once daily orally	3 months	Q10 and aspartic acid administration improved sperm kinetics, antioxidant defenses (SOD activity) and reduced nitric oxide-related oxidant species and oxidative DNA damage (*p* < 0.05).
Gvozdjáková et al., 2015 [26]	Open, uncontrolled	40	28–40	Oligoasthenoteratozoospermia	Ubiquinol 30 mg L-carnitine 440 mg Vitamin E 75 IU Vitamin C 12 mg	Twice daily during the first 3 months; once daily during the next 3 months	6 months	Increase in sperm density (*p* < 0.001). Increase in seminal CoQ10 and α-tocopherol levels and decrease in oxidative stress markers (*p* < 0.05).
Lipovac et al., 2016 [27]	Open, uncontrolled	299	20–60	Infertile men with al least one pathologic sperm analysis	Carnitine 1000 mg/day versus CoQ10 15 mg Carnitine 440 mg Arginine 250 mg Zinc 40 mg Vitamin E 120 mg Glutathione 80 mg Selenium 60 mcg	Twice daily (mono-substance) versus once daily (combination)	3 months	Improvement of all sperm parameters in both groups (*p* < 0.05), but higher relative changes in sperm density and progressive motility for the combined micronutrient treatment group.
Stenqvist et al., 2018 [12]	Randomized, double-blind, placebo-controlled	77	38	Infertile men with elevated DNA fragmentation index	CoQ10 10 mg Folic acid 100 mcg Vitamin C 30 mg Vitamin E 5 mg Vitamin B12 0.5 mcg Carnitine 750 mg Zinc 5 mg Selenium 25 mcg	Twice daily orally	6 months	Higher sperm density compared to baseline in the treatment group after antioxidant supplementation (*p* = 0.028). No differences in DNA fragmentation index in any group and between groups after treatment.
Terai et al. 2020 [28]	Prospective, randomized	31	38	Oligozoospermia and/or asthenozoospermia	CoQ10 90 mg L-Carnitine 750 mg Zinc 30 mg Astaxanthin16 mg Vitamin C 1000 mg	Three times per day	3 months	Increase in total sperm count in the supplement group after treatment (*p* = 0.04).
Arafa et al., 2020 [29]	Open, uncontrolled	148	36 (31–41)	Idiopathic male infertility and unexplained male infertility	Coq10 200 mg Vitamin A 5000 IU Vitamin C 120 mg Vitamin D3: 1200 IU Vitamin E 200 IU Vitamin K 80 µg Thiamin 3 mg Riboflavin 3.4 mg Niacin 20 mg Vitamin B6 25 mg Folate 800 µg Vitamin B12 1000 µg Biotin 600 µg Zinc 30 mg Selenium 140 µg Copper 1 mg Manganese 2 mg Chromium 120 µg L-carnitine tartrate 2000 mg L-arginine 350 mg N-acetyl l-cysteine 200 mg Lycopene 10 mg Benfotiamine 1 mg	Three capsules twice a day orally	3 months	In the idiopathic infertility group, supplementation showed a significant decrease in seminal ORP and SDF levels (*p* < 0.05). Sperm analysis revealed an improvement in all parameters investigated (*p* < 0.05), except for semen volume and sperm viability. Changes were also more evident in both the sub-categories of high ORP and high SDF. ORP levels significantly decreased in all the subjects with idiopathic infertility and in both the sub-categories of high ORP and high SDF levels (*p* < 0.05). In the unexplained male infertility group, only progressive motility significantly improved after treatment (*p* < 0.05). ORP levels decreased after treatment (*p* < 0.05).
Sadaghiani et al., 2020 [30]	Prospective, single-blinded	50	32	Astheno/oligozoospermia and cigarette smoking	CoQ10 30 mg * Zinc 8 mg * Vitamin C 100 mg * Vitamin E 12 mg * Folic acid 400 μg * Selenium 200 mg **	* Once daily ** Every other day	3 months	Increase in seminal volume (*p* = 0.032), sperm count (*p* = 0.001), sperm motility (total and progressive) (*p* = 0.001) and normal morphology (*p* = 0.003) after treatment.
Kopets et al., 2020 [31]	Randomized, double-blind, placebo-controlled	83	33	Idiopathic male infertility	Co Q10 40 Mg L-Carnitine/L-Acetyl-Carnitine 1990 Mg, L-Arginine 250 Mg Glutathione 100 Mg Zinc 7,5 Mg Vitamin B9 234 Mcg Vitamin B12 2 Mcg Selenium 50 Mcg	Once daily orally	6 months	Improvement in sperm parameters (sperm density, sperm motility and sperm morphology) (*p* < 0.001) and increase in pregnancy rate at 4 months (*p* = 0.017) in the treatment group.
Nazari et al., 2021 [32]	Open, uncontrolled	180	36 (26–40)	iOAT	Coq10 20 Mg L-Carnitine 1500 Mg Vitamin C 60 Mg Vitamin E 10 Mg Zinc 10 Mg Vitamin B9 200 µg Selenium 50 µg Vitamin B12 1 µg	Twice daily orally	3 months	Increase in sperm density (*p* = 0.004) and morphology (*p* = 0.01), but not in sperm motility

CoQ10 = coenzyme Q10; iOAT = idiopathic oligoasthenozoospermia; SOD = superoxide dismutase.

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
