# Peer review of "Coenzyme Q10 and Male Infertility: A Systematic Review"

_antioxidants, 2021, doi:10.3390/antiox10060874_

Round 1
Reviewer 1 Report
The authors are to be congratulated on the thorough search for relevant studies. I am however disappointed that the systematic search has not then precipitated systematic analysis and conclusions.
The first thing I did after first reading this article was to look on the NICE website and then Cochrane. The use of antioxidants is not specifically mentioned on NICE website. It is on Cochrane website and there is a review that generically supports the use of antioxidants but with (for Cochrane) weak evidence.
The readers of this journal will be often not doctors with an interest in male factor infertility. Therefore an explanation that seminal analysis is not a very good predictor of fertility when there are marginal results. Clearly azospermia and oligospermia are significant predictors of poor fertility. A better explanation of WHO criteria may be useful. Ultimately successful pregnancy is the outcome desired. The improvement in seminal analysis factors is only a secondary outcome.
The authors have described the inclusion criteria for the studies included but not justified these nor described the process followed to determine them. Why were studies not in English excluded? Why have they excluded gray literature as opposed to include it with caution?
The text in general consists of short descriptions of the studies and no systematic evaluation is easy to see. Tables may have been a good way of summarising these studies.
It reads to me that for each described study the emphasis of the summary of the outcome is that given. I would have liked a table that described the trend for each study for each of count, motility (a+b), morphology and total volume. For each section then a conclusion could be made based on this table.
Studies either show significant or non significant results. Lines that suggest the results are almost significant should be removed.
The great weakness of many of these studies is that they are uncontrolled. The improvement in seminal analysis may in fact be due to a generally healthier lifestyle with reduced smoking, improved diet, a more sensible exercise regime, a modification to diet/ other supplements. A quick on line search revealed many adverts for these supplements. Really we should not be promoting these supplements without clear randomised controlled trials with pregnancy outcome as the primary outcome.
The conclusions stating that CoQ10 supplements improve seminal anlysis parameters seem to be justified. The authors do state that effects on pregnancy rates need to be considered. I would want the conclusion to emphasise this more.
Reviewer 2 Report
The review entitled ”Coenzyme Q10 and Male Infertility: A Systematic Review” by Gianmaria Salvio and colleagues presents an interesting overview on a potentially relevant factor for male fertility. The study design is well described and the results are presenting the relevant studies with size, inclusion criteria, details of treatment and major outcome. However, as a review, two important aspects are missing, i.e., a nice overview on the studies and the major effects in form of a table, and a nice figure illustrating both the structure and potential biological activity of Q10 along with its potential role for male fertility. Such a table and figure would nicely complement the fine review, make it much more attractive to the readers and will lead to higher recognition in the field.
Major:
Please provide a table with the important study details (preparation of Q10, dosage, duration, country, number of participants, readouts and main effects.
Please provide a colorful representation of the biochemistry of Q10 along with its potential specific importance for male fertility and sperm health.
Minor:
Introduction: The authors highlight the importance of oxidative stress and ROS for sperm cell damage and death. Does the Se-dependent GPX4 enzyme have a role in protecting from lipid peroxidation and eventually from entering into ferroptosis as reglated pathway of spern cell death. Please comment.
Round 2
Reviewer 1 Report
This reads much better. I like the tables. I would recommend the results column has significant or non significant with p values
Author Response
Thanks for the kind reply and the valuable suggestions. P values has been added as suggested.
Reviewer 2 Report
The review has been improved strongly, especially the legibility by introducing the table and the nicely drafted figure. Congratulations on a fine contribution to an important health issue.
Author Response
Thank you for the kind response and valuable suggestions.